# A Review of Persulfate Activation by Magnetic Catalysts to Degrade Organic Contaminants: Mechanisms and Applications

**Ke Tian, Fengyin Shi, Menghan Cao, Qingzhu Zheng and Guangshan Zhang ***

College of Resource and Environment, Qingdao Engineering Research Center for Rural Environment, Qingdao Agricultural University, Qingdao 266109, China

\* Correspondence: gszhanghit@gmail.com or gszhang@qau.edu.cn; Tel./Fax: +86-532-5895-7461

**Abstract:** All kinds of refractory organic pollutants in environmental water pose a serious threat to human health and ecosystems. In recent decades, sulfate radical-based advanced oxidation processes (SR-AOPs) have attracted extensive attention in the removal of these organic pollutants due to their high redox potential and unique selectivity. This review first introduces persulfate activation by magnetic catalysts to degrade organic contaminants. We present the advances and classifications in the generation of sulfate radicals using magnetic catalysts. Subsequently, the degradation mechanisms in magnetic catalysts activated persulfate system are summarized and discussed. After an integrated presentation of magnetic catalysts in SR-AOPs, we discuss the application of persulfate activation by magnetic catalysts in the treatment of wastewater, landfill leachate, biological waste sludge, and soil containing organic pollutants. Finally, the current challenges and perspectives of magnetic catalysts that activated persulfate systems are summarized and put forward.

**Keywords:** magnetic catalysts; persulfate activation; radical pathway; non-radical pathway; mechanism and application





## 1. Introduction

In recent decades, with the rapid development of industry and agriculture, the pollution and treatment of water has become a global problem [1,2]. Some wastewater containing emerging pollutants such as antibiotics, dyes, chlorinated organic pollutants, endocrine-disrupting chemicals, and phenolics have especially caused serious environmental problems due to being irregularly discharged into the environmental medium [3–9]. These emerging pollutants have the characteristics of persistence and low biodegradability, and are difficult to be effectively removed by traditional physical or biological methods, causing serious threats to human health and ecosystems [10–12]. As an effective method of wastewater treatment, advanced oxidation processes (AOPs) play a very important role in water treatment because they can produce radicals with strong redox potential, such as hydroxyl radical (·OH), sulfate radical ($SO_4^{·-}$), etc., that lead to the degradation of pollutants [13–16].

Fenton oxidation and persulfate oxidation, as representative oxidation methods of AOPs, play an important role in the treatment of wastewater containing refractory organic pollutants. Fenton oxidation, as one of the traditional oxidation methods, has relatively high redox potential [1.8–2.7 V vs. Normal Hydrogen Electrode (NHE)] [17], which is simple to operate and does not consume energy. However, the Fenton system has some limitations, including narrow pH work range, metal ion leaching, and that hydrogen peroxide is unstable and not conducive to storage [18–20]. At present, sulfate radical-based advanced oxidation processes (SR-AOPs) have been favored by many researchers in recent decades because they have better advantages than Fenton processes. Compared with hydroxyl radical, sulfate radical has higher redox potential (2.5–3.1 V vs. NHE), longer half-life (30–40 μs vs. 20 ns), and a wider pH operating range (2~8) [21,22]. In addition, sulfate radical is selective and can preferentially attack the designated group to degrade organic pollutants

quickly and efficiently [23]. Therefore, SR-AOPs play an increasingly important role in water treatment. Generally, there are two sources of $SO_4^{\cdot-}$ production: permonosulfate (PMS) and peroxydisulfate (PDS). PMS is a compound salt ($2KHSO_5 \cdot KHSO_4 \cdot K_2SO_4$), and the trade name is oxyone [24]; PDS can be divided into sodium salt, potassium salt, and ammonium salt. Among them, sodium persulfate is commonly used as peroxydisulfate due to its stability and ease of transportation and storage [25]. PDS and PMS are not reactive at room temperature and it is hard for them to decompose by themselves [26], but can be activated by different means to produce sulfate radicals, such as transition metal ions, metal oxides, microwaves (MWs), ultraviolet light (UV), ultrasonic (US), etc., which are good persulfate activators [27–31].

However, activation of persulfate by transition metal ions causes ion leaching problems that results in secondary pollution, and could precipitate under alkaline conditions, which makes the degradation process complicated [20]. Therefore, the activation of persulfate gradually changes from a homogeneous system to a heterogeneous system. Generally, although a heterogeneous catalyst does not consume energy and the operation is simple, the recovery of catalyst is difficult and requires centrifugation or filtration [32]. Therefore, the synthesis of a magnetic catalyst becomes increasingly advantageous. On the one hand, the magnetic catalyst activates persulfate to produce an active species by electron transfer, which does not consume energy, so it can be regarded as a green catalyst. On the other hand, the magnetic catalyst can be easily recovered by the magnet without centrifugation or precipitation, which greatly saves the processing time and cost, making it an ideal persulfate activation material. For instance, Yin et al. [33] synthesized magnetic reduced graphene-$Fe_3O_4$ (rGO-$Fe_3O_4$) composite for persulfate activation by a coprecipitation method, which has good norfloxacin (NOR) removal performance; Hu et al. [34] used magnetic $Fe_3O_4$ in connection with a microwave to activate PDS to achieve enhanced degradation of *p*-nitrophenol (PNP); Cui et al. [35] synthesized a magnetic nano $Fe_3O_4$-BC catalyst via a coprecipitation method to activate PMS, demonstrating excellent catalytic stability for bisphenol A removal. Magnetic materials are easy to recover and have been paid more attention as an effective recovery method in the exploration and application of persulfate activation in related research fields.

In this paper, we first introduce different kinds of magnetic catalysts for persulfate activation to degrade refractory organic pollutants. In addition, the degradation mechanism of magnetic catalysts activated persulfate system is summarized and discussed. Then, the applications of the system in wastewater, landfill leachate, biological waste sludge, and soil containing refractory organic pollutants are introduced. Finally, the problems and future prospects of magnetic materials activated persulfate are presented, hoping to make some contributions to SR-AOPs.

## 2. Magnetic Catalysts for Persulfate Activation

### 2.1. Zero-Valent Iron (ZVI, $Fe^0$)

To avoid the secondary pollution caused by ferrous ions ($Fe^{2+}$), zero-valent iron, as a green source of $Fe^{2+}$, plays an important role in the activation of persulfate (Equation (1)) [36]. In addition, ZVI has good ferromagnetism, which is easy to recover from the degradation system and greatly reduces the treatment cost. Specifically, there are two ways for ZVI to activate persulfate. On the one hand, as a strong reducing agent, ZVI can directly reduce persulfate through an electron transfer pathway to produce sulfate radicals (Equation (2)) [37]; on the other hand, ZVI activates persulfate via slowly and smoothly releasing $Fe^{2+}$ through corrosion under aerobic or anaerobic conditions (Equations (3) and (4)) [38,39]. In addition, for ferric irons ($Fe^{3+}$) produced in the activation of persulfate by $Fe^{2+}$, the reduction of ZVI can reduce $Fe^{3+}$ to $Fe^{2+}$ (Equation (5)), thus forming a stable cycle of $Fe^{2+}/Fe^{3+}$, effectively activating persulfate to produce sulfate radicals and promoting the effective degradation

of pollutants [25,40]. Furthermore, the reaction of $Fe^{3+}$ and ZVI can effectively avoid the hydrolysis and precipitation made by the accumulation of $Fe^{3+}$ [20].

$$S_2O_8^{2-} + Fe^{2+} \rightarrow SO_4^{\cdot-} + Fe^{3+} + SO_4^{2-} \tag{1}$$

$$Fe^0 + 2S_2O_8^{2-} \rightarrow Fe^{2+} + 2SO_4^{2-} + 2SO_4^{\cdot-} \tag{2}$$

$$Fe^0 + \frac{1}{2}O_2 + H_2O \rightarrow Fe^{2+} + 2OH^- \tag{3}$$

$$Fe^0 + 2H_2O \rightarrow Fe^{2+} + 2OH^- + 2H_2 \tag{4}$$

$$Fe^0 + 2Fe^{3+} \rightarrow 3Fe^{2+} \tag{5}$$

There are many studies on the activation of persulfate by ZVI. For instance, Palharim et al. [41] explored two advanced oxidation processes: UV/persulfate system and ZVI/persulfate system, and found that ZVI can effectively activate persulfate and achieve more than 97.5% removal of propylparaben. Among them, sulfate radicals played a dominant role. Similarly, Hayat et al. [42] found that the nano-ZVI/PDS system can effectively remove 96.6% imidacloprid (IMI) while exploring the removal effect of magnetic biological carbon, nano-ferric oxide, and nano-ZVI on IMI by activating PDS, respectively. However, the difference is that ·OH is the main active species in the system.

However, these studies showed that the reduction of the pollutant removal rate was limited by the continued increase of ZVI dosage, because the excess active species would undergo a self-quenching reaction in the system, thus affecting its degradation efficiency.

## 2.2. Iron Oxide

Iron oxides have a variety of different oxidation forms, such as hematite, goethite, magnetite, pyrite, etc., and are good persulfate activator [43–46]. Because of excellent ferromagnetism, iron oxides are easy to separate and recover from the solution. The principle of persulfate activation is similar to that of zero-valent iron. Through the gradual release of $Fe^{2+}$ from the surface of iron oxide into the solution, persulfate can be activated to produce active species and then participate in redox reaction [34]. In recent decades, $Fe_3O_4$ has a good application prospect in activated persulfate. For example, Zhao et al. [47] synthesized magnetic $Fe_3O_4$ by a coprecipitation method, and the $Fe_3O_4$/PDS system can completely remove p-nitroaniline at neutral pH. Yan et al. [48] synthesized magnetic $Fe_3O_4$ by a reverse coprecipitation method, which completely removed 0.06 mmol/L sulfamethoxine in 15 min. The degradation process produced harmless intermediates, which can be considered as a green persulfate oxidant.

In addition, other iron oxides can also be used as excellent activators of persulfate. Hussain et al. [49] synthesized a magnetic $BiFeO_3$ nano-catalyst by a sol-gel method, which can effectively activate persulfate to remove aniline. Sulfate radicals and hydroxyl radicals are the main active species in the system. In addition, the repeated experiments indicated that the material has high stability, which can still remove 93.3% aniline in the solution after five recycling cycles. Guan et al. [50] synthesized magnetic copper ferrite ($CuFe_2O_4$) by a sol-gel method to activate persulfate, and the system can remove more than 98% of atrazine within 15 min.

Iron oxides come in various forms and they are easily recovered because of their magnetic properties. In addition, they can also be used as a green catalytic material, which can be further explored in the activation of persulfate.

## 2.3. Nickel-Cobalt Bimetallic Catalyst

Transition metal cobalt ion is one of the effective activators of PDS, but due to its high leaching concentration, it is easy to cause secondary pollution, which is harmful to the ecosystem and requires further treatment. In recent years, as a low-cost, magnetically separable catalyst, nickel-cobalt bimetallic catalyst has a certain application prospect in the activation of persulfate. Tian et al. [51] prepared dandelion $NiCo_2O_4$ microspheres by

the hydrothermal method, which has a good mesoporous structure and can effectively activate persulfate to degrade humic acid (HA). The introduction of Ni effectively reduces the toxicity of Co and can be considered as a green catalyst for environmental protection. Wu et al. [52] synthesized a $NiCo_2O_4$ catalyst with a porous network structure to activate PMS to degrade tetracycline (TC) and bisphenol A (BPA). The high catalytic performance can be attributed to the covalent conversion between $Ni^{2+}/Ni^{3+}$ and $Co^{2+}/Co^{3+}$ (Figure 1).

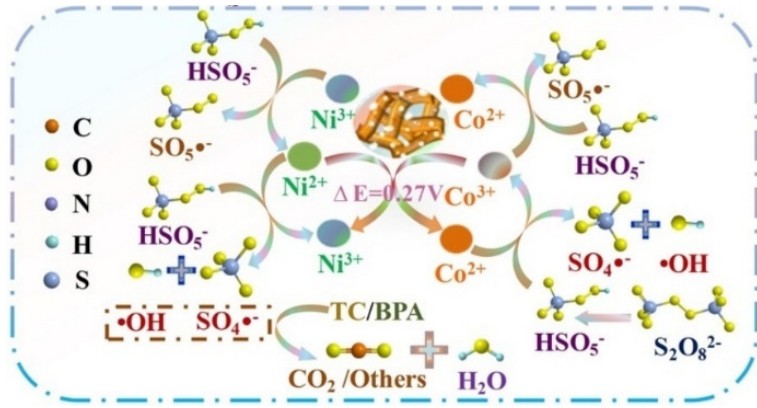

**Figure 1.** The recycle of $Ni^{2+}/Ni^{3+}$ and $Co^{2+}/Co^{3+}$ and the removal mechanism of TC and BPA in the $NiCo_2O_4$/PMS system [52].

In addition, as a magnetic carrier, nickel foam can be easily recovered from the reaction solution. Hu et al. [53] used a microwave-assisted method to load $Co_3O_4$-$Bi_2O_3$ nanoparticles on the nickel foam carrier to degrade BPA via persulfate activation. The excellent catalytic activity came from the electron transfer between $Co^{2+}/Co^{3+}$, $Bi^{3+}/Bi^{5+}$ and $Ni^{2+}/Ni^{3+}$, realizing the continuous regeneration of active species, and 95.6% of BPA was removed within 30 min.

*2.4. Supported Magnetic Catalyst*

However, a common problem with the magnetic materials mentioned above is that they are easy to aggregate [54]. Therefore, researchers are increasingly interested in loading magnetic materials onto various forms of biochar carriers to produce magnetic biocarbon materials [32,55,56]. On the one hand, the aggregation of nanomaterials can be reduced; on the other hand, the advantages of carbonaceous materials can be fully exploited, and the synergistic effect of the nanomaterials and carbonaceous materials can be well applied to the removal of organic pollutants [57]. There are abundant sources of carbon materials in nature, many of which exist in the form of wastes [54]. The secondary utilization of carbon through high-temperature calcination or pyrolysis can effectively realize the resource utilization of wastes [58].

Abundant forms of carbon provide a variety of carrier forms for magnetic materials, so kinds of magnetic carbon materials with high efficiency in activating persulfate were developed. Fu et al. [59] used *Myriophyllum aquaticum* composites as a carbon source, and prepared a $Fe_3O_4$-porous biological carbon ($Fe_3O_4$-MC) catalyst with a highly graphitized structure, multi-porous structure (homogeneous distribution of mesopores and micropores), and strong magnetic properties through high temperature pyrolysis. At the pyrolysis temperature of 800 °C, $Fe_3O_4$-MC can effectively activate PMS and completely remove *p*-hydroxybenzoic acid (HBA) within 30 min. Yin et al. [33] prepared a nanosheet reduced rGO-$Fe_3O_4$ catalyst by a coprecipitation method. After four recycling cycles, the rGO-$Fe_3O_4$/PDS system can still remove 74.99% of NOR (Figure 2a), and the XRD characteristic diffraction peaks before and after the reaction do not change significantly (Figure 2b), indicating that the material has good stability. Zhang et al. [56] used ferrocene and carbon nanofibers (CNFs) as precursors, and prepared carbon-encapsulated $Fe_3O_4$ ($Fe_3O_4$@C/CNFs) magnetic materials on carbon nanofibers. The CNF carrier effectively

avoids the agglomeration of $Fe_3O_4$, and the leaching amount of iron ions after five recycling cycles is very low, which is about 1/20 of the leaching amount of iron ions in commercial $Fe_3O_4$ under the same conditions.

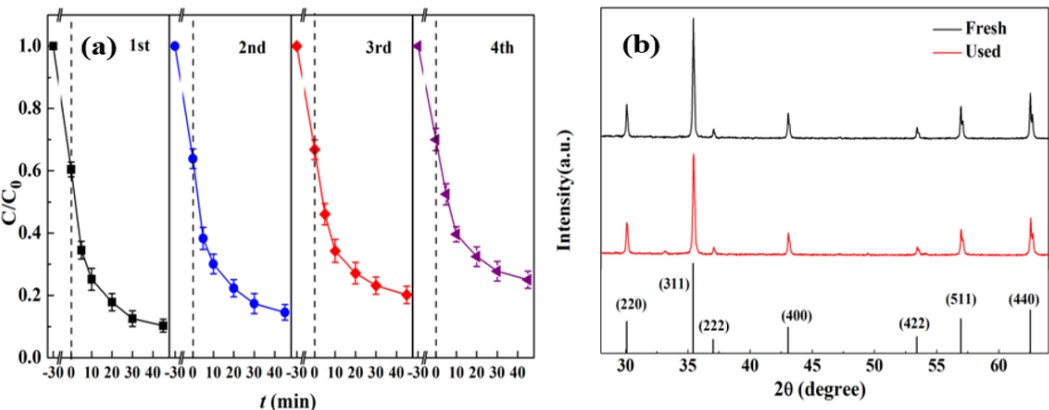

**Figure 2.** Recycling experiment of rGO-$Fe_3O_4$ (**a**) and XRD patterns of the fresh and used rGO-$Fe_3O_4$ (**b**) [33].

Generally, iron, cobalt, and nickel are the three main transition metals in magnetic materials, and the existence of a carrier is beneficial to the aggregation of nanoparticles. In addition, magnetic materials have bright prospects in persulfate activation due to their excellent recovery ability, simple operation, and energy saving.

## 3. Mechanism for Persulfate Activation by Magnetic Catalysts

In SR-AOPs, the reactive oxygen species (ROS) produced in persulfate systems vary with catalysts and target pollutants. According to the nature of active species, the activation mechanism of persulfate can be divided into a radical pathway and non-radical pathway. The main radical pathways are sulfate radicals, hydroxyl radicals, and superoxide anion radicals; non-radical pathways mainly include electron transfer pathways, singlet oxygen, and complexes formed by catalytic materials and persulfate adsorbed on them during the reaction. Therefore, it is important to identify these active species during the reaction first.

### 3.1. Identification of Reactive Oxygen Species

Two methods are commonly used to explore ROS in the system: (1). quenching experiments; (2). electron paramagnetic resonance (EPR) or electron spin resonance (ESR). In the quenching experiment, a certain amount of quench agents are added to the system to react with the corresponding active species, and the existence of the active species is judged according to the inhibition effect on the degradation of targeted pollutants. The identification of the active species is determined by the reaction rate constant ($k_{obs}$) between the quenching agent and the active species (Table 1). For common quenching agents, both methanol (MeOH) and ethanol (EtOH) can be used to identify $SO_4^{\cdot-}$ and $\cdot OH$, and tert-butanol (TBA) can be used to identify $\cdot OH$ [60–64]. Since MeOH, EtOH, and TBA are hydrophilic and tend to react with active species in solution, potassium iodide (KI) and phenol are used to identify $SO_4^{\cdot-}$ and $\cdot OH$ on the surface of materials, and nitrobenzene (NB) is used to identify $\cdot OH$ on the surface of catalyst materials [59,61,65–69]. Para-benzoquinone (*p*-BQ) is used to identify $O_2^{\cdot-}$ [66,70]. Both furfuryl alcohol (FFA), L-Listidine, and sodium azide ($NaN_3$) can used to selectively react with $^1O_2$ [66,71–75]. In addition, the reduction of Zeta potential could affect the electrostatic binding between solute and suspended solid in solution, and thus hinder the electronic transfer of organic pollutants to the catalyst to a certain extent. However, the presence of $NaClO_4$ will reduce the Zeta potential of the catalyst [59,68,76]. Therefore, adding $NaClO_4$ to the reaction system can identify $e^-$.

**Table 1.** The reaction rate constants of commonly used quenching agents with active species.

| Quench Agents | Targeted ROS | Reaction Rate Constants $(M^{-1} \cdot s^{-1})$ | Identification of ROS | Location of ROS | Ref. |
|---|---|---|---|---|---|
| MeOH | $SO_4^{\cdot -}$ | $3.2 \times 10^6$ | $SO_4^{\cdot -}$, $\cdot OH$ | In the solution | [60–62] |
| | $\cdot OH$ | $9.7 \times 10^8$ | | | |
| TBA | $SO_4^{\cdot -}$ | $4–9.1 \times 10^5$ | $\cdot OH$ | In the solution | |
| | $\cdot OH$ | $3.8–7.6 \times 10^8$ | | | |
| EtOH | $SO_4^{\cdot -}$ | $1.6–7.7 \times 10^7$ | $SO_4^{\cdot -}$, $\cdot OH$ | In the solution | [63,64] |
| | $\cdot OH$ | $1.2–2.8 \times 10^8$ | | | |
| Phenol | $SO_4^{\cdot -}$ | $8.8 \times 10^9$ | $SO_4^{\cdot -}$, $\cdot OH$ | Catalyst surface | [65,66] |
| | $\cdot OH$ | $6.6 \times 10^9$ | | | |
| KI | $SO_4^{\cdot -}$ | - | $SO_4^{\cdot -}$, $\cdot OH$ | Catalyst surface | [61,67] |
| | $\cdot OH$ | - | | | |
| NB | $SO_4^{\cdot -}$ | $<10^6$ | $\cdot OH$ | Catalyst surface | [59,68,69] |
| | $\cdot OH$ | $3.9 \times 10^9$ | | | |
| *p*-BQ | $O_2^{\cdot -}$ | $1.0 \times 10^9$ | $O_2^{\cdot -}$ | Catalyst surface | [66,70] |
| L-listidine | | $3.2 \times 10^7$ | | | [71,72] |
| FFA | $^1O_2$ | $1.2 \times 10^8$ | $^1O_2$ | - | [73,74] |
| NaN$_3$ | | $1.0 \times 10^9$ | | | [66,75] |
| NaClO$_4$ | $e^-$ | - | $e^-$ | - | [59,68,76] |

Another method is electron paramagnetic resonance (EPR) or electron spin resonance (ESR). Direct detection with EPR or ESR is difficult due to the relatively short life span of active species. Therefore, spin capture agents are added to combine with active species to form relatively stable spin admixtures, which are converted into paramagnetic species easily measured by EPR or ESR [77]. For $SO_4^{\cdot -}$, $\cdot OH$ and $O_2^{\cdot -}$, 5,5-dimethyl-1-pyrroline-1-oxide (DMPO) is generally selected as the optional trapping agent, thus forming DMPO-$SO_4^{\cdot -}$, DMPO-$\cdot OH$, and DMPO-$O_2^{\cdot -}$ spin admixtures [56,59,78]; $^1O_2$ can be captured by 2,2,6,6-tetramethyl-4-piperidine (TEMP) to form the spin adduct TEMP-$^1O_2$ [79]. After the adducts are obtained, hyphenate splitting constants (HPC) are calculated using specialized software related to EPR technology, and these constants are then compared with databases or the literature to identify active species [77].

As shown in Figure 3, for the DMPO-$\cdot OH$ adduct, it generally presents a quaternion characteristic peak of 1:2:2:1 (HPC:$\alpha_N$ = 14.9 G, $\alpha_{\beta-H}$ = 14.9 G) (Figure 3a) [77,80]; for DMPO-$SO_4^{\cdot -}$ admixtures, the characteristic peaks generally show a six-component characteristic peak of 1:1:1:1:1:1 (HPC:$\alpha_N$ = 13.51 G, $\alpha_{\beta-H}$ = 9.93 G, $\alpha_{\gamma-H1}$ = 1.34 G, $\alpha_{\gamma-H2}$ = 0.88 G) (Figure 3a) [77,80]. It is worth noting that some studies have found that quenching experiments proved the existence of $SO_4^{\cdot -}$, $\cdot OH$, but no corresponding characteristic peak was found in EPR. However, a seven-element peak of 1:2:1:2:1:2:1 appeared due to the oxidation of DMPO by oxidizing species ($SO_4^{\cdot -}$, $\cdot OH$) into a special admixture DMPO-X, which can also explain the existence of $SO_4^{\cdot -}$, $\cdot OH$ (Figure 3b) [59,81]. For DMPO-$O_2^{\cdot -}$ admixtures, the characteristic peaks generally show a six-component characteristic peak of 1:1:1:1:1:1 (HPC:$\alpha_N$ = 14.3 G, $\alpha_{\beta-H}$ = 11.2 G, $\alpha_{\gamma-H1}$ = 1.3 G) (Figure 3c) [77,82]. For the TEMP-$^1O_2$ adduct, it generally presents a ternary characteristic peak of 1:1:1 (HPC:$\alpha_N$ = 16.3 G) (Figure 3d) [77,82].

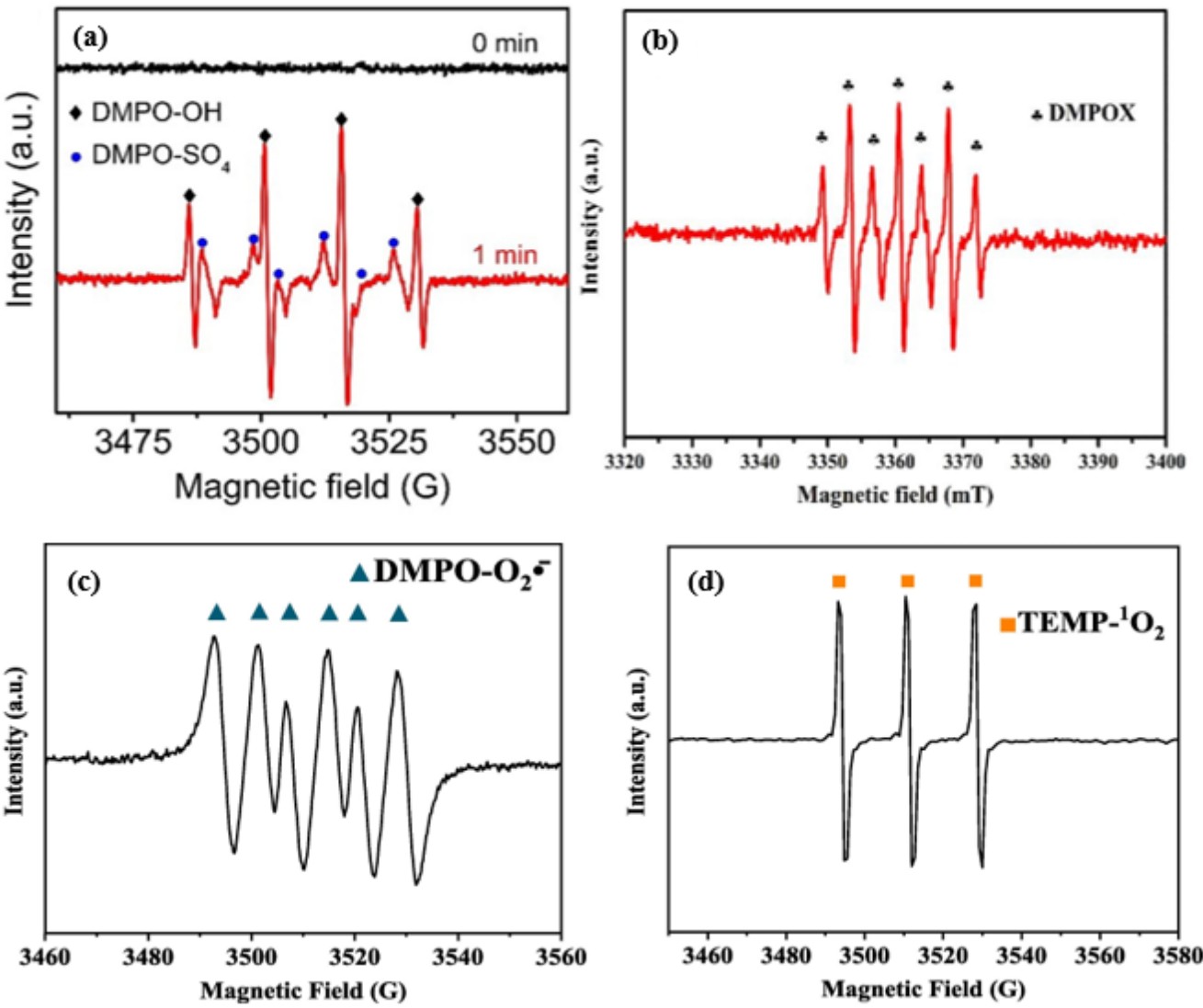

**Figure 3.** EPR characteristic spectrum of (**a**) DMPO-SO$_4$$^{\cdot-}$ and DMPO-·OH; (**b**) DMPO-X; (**c**) DMPO-O$_2$$^{\cdot-}$; and (**d**) TEMP-$^1$O$_2$ [80–82].

In general, the combination of the quenching experiment and EPR/ESR can strongly explain the active species in the system and provide strong support for the activation mechanism of persulfate. Therefore, when exploring the activation mechanism in the reaction process, it is necessary to combine the above two kinds of active species identification methods as far as possible, to make the article more scientific and rigorous.

### 3.2. Radical Pathway and Non-Radical Pathway

In persulfate systems, different catalysts play different roles with persulfates and have different activation mechanisms. Some are based on the radical pathway, some are based on the non-radical pathway, and some combine the radical pathway with the non-radical pathway. Here, the following section will be introduced, respectively, and some active species involved in the reaction process were summarized in Table 2 [33–35,37,53,55,59,79,83–93].

**Table 2.** Active species involved in the reaction process.

| Reaction Pathway | System | Targeted Pollutant | pH | Quenching Agent (Degradation Rate after Inhibition, Control Degradation Rate) | EPR Signal | Activation Mechanism | Ref. |
|---|---|---|---|---|---|---|---|
| Radical pathway | $Fe_3O_4$-BC/PMS | BPA | 3.0 | TBA (84.5%, 100%) EtOH (47.6%, 100%) | DMPO-X | $SO_4^{\cdot-}$ (major), ·OH | [35] |
| | nZVI-Omt/PDS | SMZ | 6.8 | TBA (73%, 97%) MeOH (40%, 97%) | DMPO-·OH DMPO-$SO_4^{\cdot-}$ | $SO_4^{\cdot-}$, ·OH | [37] |
| | nZVI-BC/PDS | TC | 5.0 | TBA (75.95%, 87.58%) EtOH (32.65%, 87.58%) | DMPO-·OH DMPO-$SO_4^{\cdot-}$ | $SO_4^{\cdot-}$ (major), ·OH | [83] |
| | $Co_3O_4$-$CeO_2$/PDS | TC | 7 | TBA (52%, 79%) MeOH (41%, 79%) | DMPO-·OH DMPO-$SO_4^{\cdot-}$ | $SO_4^{\cdot-}$, ·OH(major) | [84] |
| | $Co_3O_4$-$Bi_2O_3$@NF/PMS | BPA | 3.4 | TBA (~95%, 95.6%) MeOH (~95%, 95.6%) KI (~8%, 95.6%) | DMPO-·OH DMPO-$SO_4^{\cdot-}$ | $SO_4^{\cdot-}{}_{surface}$, ·$OH_{surface}$ | [53] |
| | $Ag_{0.4}$-$BiFeO_3$/PDS | TC | 4.5 | TBA (~48%, 91%) MeOH (~52%, 91%) | DMPO-·OH DMPO-$SO_4^{\cdot-}$ | $SO_4^{\cdot-}$, ·OH(major) | [85] |
| | $Fe_3O_4$@$CuO_x$/PDS | sulfadiazine (SDZ) | 7 | TBA (58.6%, 95%) MeOH (18.2%, 95%) | DMPO-·OH DMPO-$SO_4^{\cdot-}$ | $SO_4^{\cdot-}$ (major), ·OH | [86] |
| | $Fe_3O_4$/hf-CuO/PDS | 4-aminobenzenesulfonic acid (4-ABS) | 7 | TBA (45.7%, 90%) MeOH (4.3%, 90%) | DMPO-·OH DMPO-$SO_4^{\cdot-}$ | $SO_4^{\cdot-}$, ·OH | [87] |
| | $Fe_3O_4$/MW/PDS | PNP | 3.4 | TBA (76.2%, 98.2%) MeOH (29.3%, 98.2%) | DMPO-·OH DMPO-$SO_4^{\cdot-}$ | $SO_4^{\cdot-}$ (major), ·OH | [34] |
| Non-radical pathway | RC/CNTs/$Fe_3O_4$ NPs/PDS | BPA | 6.07 | TBA (~100%, 100%) EtOH (~100%, 100%) FFA (~60%, 100%) | TEMP-$^1O_2$ | $^1O_2$, electron transfer, catalyst-PDS * | [55] |
| | rGO-$Fe_3O_4$/PDS | NOR | 6.47 | TBA (~82%, 89.6%) EtOH (~73%, 89.6%) FFA (~50%, 89.6%) | - | $^1O_2$, electron transfer | [33] |
| | Ni-NiO/PDS | 4-CP | 7.0 | MeOH (~80%, 100%) (The removal of FFA is lower than 10%) | - | Electron transfer | [88] |
| | $Co_3O_4$@NCNTs/PDS | Orange G (OG) | 7.0 | TBA (90.1%, 100%) EtOH (~90%, 100%) FFA (17.3%, 100%) p-BQ (~70%, 100%) | DMPO-·OH DMPO-$SO_4^{\cdot-}$ DMPO-$O_2^{\cdot-}$ TEMP-$^1O_2$(major) | $^1O_2$(major), electron transfer, catalyst-PDS * | [89] |
| | UBC-x/PMS | BPA | 6.84 | TBA (~100%, 100%) EtOH (~100%, 100%) FFA (75%, 100%) | DMPO-·OH DMPO-$SO_4^{\cdot-}$ DMPO-$O_2^{\cdot-}$ TEMP-$^1O_2$ | $^1O_2$(major) | [90] |

**Table 2.** *Cont.*

| Reaction Pathway | System | Targeted Pollutant | pH | Quenching Agent (Degradation Rate after Inhibition, Control Degradation Rate) | EPR Signal | Activation Mechanism | Ref. |
|---|---|---|---|---|---|---|---|
| Radical pathway and non-radical pathway | $Fe_3O_4$/MC/PMS | HBA | - | Phenol (10%, 100%)<br>NB (88%, 100%)<br>*p*-BQ (67%, 100%)<br>NaClO4 (70%, 100%) | DMPO-X | $SO_4^{\cdot-}$-surface, $\cdot OH$, $O_2^{\cdot-}$, electron transfer | [59] |
| | Fe-N-BC/PDS | acid orange (AO7) | 7.0 | TBA (56.9%, 98.2%)<br>MeOH (44.6%, 98.2%)<br>KI (~30%, 98.2%)<br>FFA (~68%, 98.2%)<br>BQ (~35%, 98.2%) | DMPO-$\cdot OH$<br>DMPO-$SO_4^{\cdot-}$<br>DMPO-$O_2^{\cdot-}$<br>TEMP-$^1O_2$ | $SO_4^{\cdot-}$, $\cdot OH$, $O_2^{\cdot-}$, $^1O_2$, electron transfer, catalyst-PDS * | [79] |
| | FeCN*x*/PMS | BPA | 6.5 | MeOH (~90%, 94%)<br>KI (~42%, 94%)<br>NaN$_3$ (~16%, 94%) | DMPO-$\cdot OH$<br>DMPO-$SO_4^{\cdot-}$<br>TEMP-$^1O_2$ | $SO_4^{\cdot-}$-surface, $\cdot OH$-surface, $^1O_2$, electron transfer, catalyst-PMS * | [91] |
| | $Fe_3O_4$/ $CoCO_3$/rGO/PMS | Rhodamine B (RhB) | 7.0 | TBA (72.5%, 98.69%)<br>EtOH (10.29%, 98.69%)<br>FFA (complete inhibition) | - | $SO_4^{\cdot-}$ (major), $^1O_2$(major), $\cdot OH$ | [92] |
| | nZVI@NBC/PDS | BPA | 7.0 | TBA (77.3%, 100%)<br>EtOH (74.1%, 100%)<br>FFA (12%, 100%) | DMPO-$\cdot OH$<br>DMPO-$SO_4^{\cdot-}$<br>TEMP-$^1O_2$ | $SO_4^{\cdot-}$, $\cdot OH$, $^1O_2$ | [93] |

Note: "*" represents the complex formed between the catalyst and the oxidant adsorbed on its surface.

3.2.1. Radical Pathway

In persulfate systems, persulfates are activated by magnetic catalysts to produce $SO_4^{\cdot-}$, ·OH, and $O_2^{\cdot-}$ (Equations (6)–(12)) [10,94,95]. Due to its strong redox ability, $SO_4^{\cdot-}$, ·OH and $O_2^{\cdot-}$ can act with target pollutants and mineralize them into small molecules.

In most systems, $SO_4^{\cdot-}$ was the main active species. For example, Cui et al. [35] used a coprecipitate method to synthesize a nano $Fe_3O_4$-BC catalyst to activate persulfate, and EtOH and TBA were used to quench $SO_4^{\cdot-}$ and ·OH. The inhibitory effect of EtOH on BPA was significantly higher than that of TBA, and an EPR experiment showed DMPO-X's characteristic peak, indicating that DMPO was oxidized by $SO_4^{\cdot-}$ and ·OH, therefore, the system was dominated by a $SO_4^{\cdot-}$-based radical system. Shao et al. [83] synthesized $Fe_3O_4$ loaded by BC ($Fe_3O_4$-BC) to activate PDS to degrade tetracycline. EtOH and TBA were chosen to quench $SO_4^{\cdot-}$ and ·OH. The results showed that the inhibition effect of EtOH on TC was significantly higher than that of TBA. DMPO-$SO_4^{\cdot-}$ and DMPO-·OH signals were also detected in the EPR experiment, therefore, the system was also a $SO_4^{\cdot-}$ dominated radical system.

In addition, ·OH played a major role in some systems, which may be related to the pH of the reaction or the nature of targeted pollutant [84,85]. Ouyang et al. [85] prepared a silver-doped bismuth ferrite composite ($Ag_{0.4}$-$BiFeO_3$) by a sol-gel method, which was used to activate PMS and degrade TC. DMPO-$SO_4^{\cdot-}$ and DMPO-·OH signals were detected by ESR, and DMPO-·OH signals were stronger than DMPO-$SO_4^{\cdot-}$ signals. To further demonstrate the existence of active species, methanol and tert-butanol were used to quench $SO_4^{\cdot-}$ and ·OH, and the inhibition effect of tert-butanol on TC in the system was slightly weaker than methanol. Therefore, both $SO_4^{\cdot-}$ and ·OH in an $Ag_{0.4}$-$BiFeO_3$/PMS system are involved in degradation reactions. Specifically, low-valent transition metal ions activate PMS to produce sulfate radicals, then sulfate radicals react with an $H_2O$ molecule or hydroxide ion in a solution to produce ·OH, and ·OH is the main one in the system.

However, not all persulfate systems can be investigated for the presence of radicals by quenching and EPR experiments. In some persulfate systems, other methods have been used by some researchers to demonstrate radical pathway. Guan et al. [50] synthesized copper ferrate catalyst ($CuFe_2O_4$) to activate PMS to degrade atrazine by a sol-gel method, assuming only ·OH exists in the reaction system. Therefore, the removal of NB and ATZ by $CuFe_2O_4$/PMS system was compared, and the contribution of ·OH on the catalyst surface was investigated according to the ratio of quasi-first-order rate constant of reaction kinetics of each system during the degradation process. The results showed that the actual value of the ratio of quasi-first-order rate constants ($k_{ATZ}/k_{NB}$ = 6.4) is much higher than the theoretical value (0.5~1), which proves that $SO_4^{\cdot-}$ and ·OH are the main active species on the catalyst surface.

Magnetic catalyst activates persulfate to produce radicals such as $SO_4^{\cdot-}$, ·OH, and $O_2^{\cdot-}$ with strong oxidation to attack pollutants, and pollutants are broken down into small molecules of inorganic oxide, $CO_2$ and $H_2O$ in a short time. In general, the radical pathway is an effective way to remove pollutants by AOPs.

$$Fe^{2+}/Co^{2+}/Ni^+ + S_2O_8^{2-} \rightarrow SO_4^{\cdot-} + SO_4^{2-} + Fe^{3+}/Co^{3+}/Ni^{2+} \tag{6}$$

$$Fe^{2+}/Co^{2+}/Ni^+ + HSO_5^- \rightarrow SO_4^{\cdot-} + OH^- + Fe^{3+}/Co^{3+}/Ni^{2+} \tag{7}$$

$$Ag^+ + S_2O_8^{2-} \rightarrow SO_4^{\cdot-} + SO_4^{2-} + Ag^{2+} \tag{8}$$

$$SO_4^{\cdot-} + OH^- \rightarrow \cdot OH + SO_4^{2-} \tag{9}$$

$$SO_4^{\cdot-} + H_2O \rightarrow \cdot OH + SO_4^{2-} + H^+ \tag{10}$$

$$S_2O_8^{2-} + 2H_2O \rightarrow 2SO_4^{2-} + HO_2^- + 3H^+ \tag{11}$$

$$S_2O_8^{2-} + HO_2^- \rightarrow SO_4^{2-} + SO_4^{\cdot-} + O_2^{\cdot-} + H^+ \tag{12}$$

### 3.2.2. Non-Radical Pathway

However, in other persulfate systems, radicals play a very small role, and the removal of pollutants is mainly induced by the non-radical pathway including singlet oxygen, electron transfer, and complex. Generally, there are three production sources of $^1O_2$: (1). Transformation of superoxide anion radicals (Equations (13)–(15)) [82,96]; (2). The decomposition of persulfates occurs directly (Equations (16)–(18)) [86,92,97,98]; and (3). Transformation of functional groups. For example, because of the existence of the O-O bond, persulfates are easily polarized by catalytic materials with polar structure to produce $^1O_2$, and $^1O_2$ can also be induced by the presence of C=O and graphite N according to different catalyst materials, especially the magnetic catalyst with carbon material as the carrier [99,100]. Studies have shown that $^1O_2$ owns highly selectivity and tends to attack electron-rich organic pollutants such as phenols [101]. Electron transfer pathway also plays an important role in the degradation of pollutants. Generally, catalysts are good electron transfer agents (electron mediators), can transfer electrons from adsorbed organic pollutants (electron donors) to persulfate (electron acceptors), promoting redox reactions [102]. The formation of complexes generally takes place on magnetic catalysts supported by carbon materials. That is, carbon with a positive charge forms a stable complex with high redox capacity due to the strong electrostatic binding force between it and PDS/PMS (catalyst-PDS*, catalyst-PMS*) (Equation (19)) [55,91].

$$O_2^{\cdot -} + \cdot OH \rightarrow {}^1O_2 + OH^- \tag{13}$$

$$2O_2^{\cdot -} + 2H^+ \rightarrow H_2O_2 + {}^1O_2 \tag{14}$$

$$2O_2^{\cdot -} + 2H_2O \rightarrow 2OH^- + {}^1O_2 + H_2O_2 \tag{15}$$

$$HSO_5^- \rightarrow SO_5^{2-} + H^+ \tag{16}$$

$$SO_5^{2-} + HSO_5^- \rightarrow {}^1O_2 + HSO_4^- + SO_4^{2-} \tag{17}$$

$$S_2O_8^{2-} + \cdot OH \rightarrow SO_4^{2-} + SO_4^{\cdot -} + H^+ + \frac{1}{2}{}^1O_2 \tag{18}$$

$$Catalyst + PDS/PMS \rightarrow catalyst - PDS^*/PMS^* \tag{19}$$

Different from the radical pathway, the activation mechanisms explored in the following systems are mainly induced by the non-radical pathway. For instance, Yin et al. [33] synthesized the rGO-Fe$_3$O$_4$ composite catalyst to activate PDS via a coprecipitation method. For the reaction mechanism of NOR, the author found that $^1O_2$ is the main active species relative to $SO_4^{\cdot -}$ and $\cdot OH$ by quenching experiment. The timing of the current analysis showed that the current jump was significant only in the presence of the contaminant NOR, catalyst rGO-Fe$_3$O$_4$, and oxidant PDS. This phenomenon demonstrates the existence of an electron transfer pathway from pollutant to catalyst and then to oxidant. Thus, the coexistence of singlet oxygen and electron transfer promoted the degradation of NOR. Kim et al. [88] synthesized a nanocomposite catalyst of nickel metal and nickel oxide (Ni-NiO) by a sol-gel method to activate PDS for the degradation of 4-chlorophenol (4-CP). EPR could not detect the signal of radicals, ans methanol was used to quench the possible active species, and there was almost no inhibition on the removal of 4-CP, proving that the reaction was a non-radical pathway. It is worth noting that the Ni-NiO/PDS system has less than 10% removal effect on FFA, which also excludes the effect of $^1O_2$. Linear sweep voltammetry (LSV) and PDS decomposition experiments showed that Ni-NiO catalyst acts as electron transfer medium and promotes electron transfer from pollutant to PDS. Furthermore, based on the previous studies, the authors hypothesized that Ni-NiO forms reactive complexes with PDS adsorbed on its surface, leading to activation of PDS and oxidation of contaminants in the system.

Similarly, through a series of electrochemical signal experiments, Lai' team also found that the removal of pollutants in the persulfate system was based on the generation of complex as the main oxidative species [55]. Specifically, the team synthesized magnetic renewable fiber-loaded carbon nanotubes/ferric oxide nanocomposites (RC/CNTs/Fe$_3$O$_4$

NPs) to activate PDS and degrade BPA. Two electrochemical characterization methods, electrochemical impedance spectroscopy (EIS) and Tafel polarization curve, verified the existence of electron transfer pathway (Figure 4a,b). The existence of the complex was also proven by electrochemical experiments as well. First, catalyst and oxidizer combined to form catalyst-PDS*, and the composite material transfers some electrons to PDS, which leads to the increase of oxidation potential of catalyst-PDS*. When a new equilibrium is reached, BPA was added, and catalyst-PDS* accepts electrons from BPA, thus reducing the oxidation potential. Then, the oxidation potential of catalyst-PDS* increased when the content of BPA decreased with degradation. Finally, when the content of BPA decreased with degradation, the oxidation potential of catalyst-PDS* rose again. Here, when BPA was added again, the oxidation potential of catalyst-PDS* decreased first and then rose again, which proved the existence of catalyst-PDS* complex (Figure 4c). In addition, the quenching and EPR experiments showed that $^1O_2$ was the main oxidizing species. Hence, $^1O_2$, together with the electron transfer pathway and the formation of the complex promoted the degradation of BPA (Figure 4d).

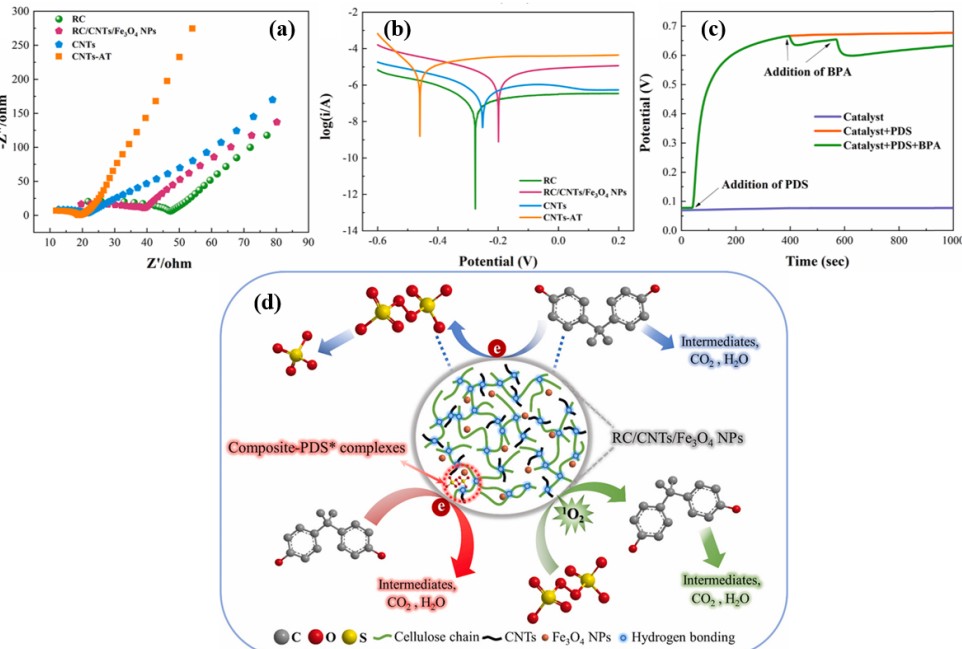

**Figure 4.** Electrochemical impedance spectroscopy. (**a**) Tafel polarization curve; (**b**) the open circuit potential curve (**c**) and schematic diagram of the mechanism (**d**) of the BPA removal in the RC/CNTs/Fe$_3$O$_4$/PDS system [55].

Reaction mechanisms dominated by the non-radical pathway are gaining increasing attention in persulfate-based advanced oxidation processes because of their ability to overcome some of the limitations of radical activation (such as short half-lives). Although the non-radical pathway does not directly rely on strong oxidation to attack pollutants (except $^1O_2$), it also promotes the redox reaction by forming a complex with strong oxidation or by electron transfer in the reaction system, causing the degradation of pollutants. It is worth noting that the non-radical pathway is not a single one at work, but may involve two or more pathways, which is worth noting when exploring the activation mechanism of persulfate.

### 3.2.3. The Synergetic Radical Pathway and Non-Radical Pathway

Under the activation of some composite catalyst, the method of persulfate degradation of pollutants diversification involves the radical approaches containing sulfate radicals, hydroxyl radicals, superoxide anion radicals, and is covered with a singlet oxygen, electron

transfer, and the complex of the non-radical ways, which jointly promote the efficient degradation of pollutants.

For reactions involving both the radical pathway and the non-radical pathway, researchers have done many experiments to demonstrate the existence of active species (including quenching experiments, EPR experiments, electrochemical experiments, etc.). For instance, Fu et al. [59] synthesized a $Fe_3O_4$-graphitized porous biological carbon composite catalyst ($Fe_3O_4$/MC) to activate PMS to degrade $p$-hydroxybenzoic acid. In the quenching experiment, phenol, NB, and BQ can inhibit HBA degradation, proving that $SO_4^{\cdot-}$, $\cdot OH$ and $O_2^{\cdot-}$ are involved in the reaction, and mainly induced by $SO_4^{\cdot-}$ and $\cdot OH$. The presence of $NaClO_4$ also inhibited the removal of HBA, and combined with EIS and LSV, it was proven that there was a non-radical pathway supplemented by electron transfer pathway in the system. The combination of radical and non-radical pathway promotes the degradation of pollutants. Miao et al. [91] synthesized a Fe and N co-doped carbon-based catalyst (FeCN$x$) to activate PMS, and explored the mechanism of this system in the degradation of BPA. The presence of $^1O_2$ and surface-bound $SO_4^{\cdot-}$ and $\cdot OH$ was confirmed by the quenching experiment and EPR detection. In addition, FeCN$x$ and the PMS adsorbed on the surface of the catalyst and formed a stable and highly active complex (catalyst-PMS*). The electron transfer from BPA to catalyst-PMS* was mediated by the catalyst, triggering the formation of the complex and promoting the degradation of BPA (Figure 5). Dung et al. [92] synthesized a $Fe_3O_4$/CoCO$_3$/rGO magnetic catalyst by a one-step solvothermal method to activate PMS and degrade rhodamine dye. Both $SO_4^{\cdot-}$ and $^1O_2$ contribute greatly to the degradation of rhodamine by the radical and non-radical pathways. In addition, the combination of these two ways also has a good removal effect for other dyes, reflecting the width of its application.

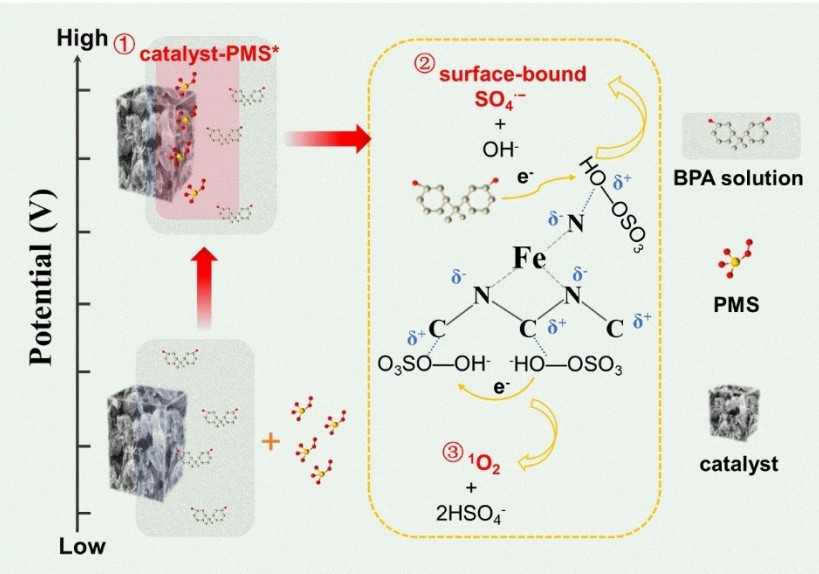

**Figure 5.** Mechanism of removal of BPA by FeCN$x$/PMS system [91].

The synergistic effect of the radical and non-free radical pathways can effectively promote the degradation of pollutants, especially some structurally stable pollutants (such as phenolic substances). The above studies indicate that the combination of the radical and non-radical pathways can extend the application range of the system, and it has a good application prospect for the treatment of practical wastewater with complex components.

## 4. The Application of the Persulfate/Magnetic Catalyst System

Due to the rapid development of industry and agriculture, refractory organic pollutants not only exist in wastewater, but also in landfill leachate, biological waste sludge, and

soil. A persulfate/magnetic catalyst system has some applications in these practical fields (Table 3), and some specific cases are described in the following sections.

**Table 3.** The application of persulfate/magnetic catalyst system in actual wastewater, landfill leachate, biological waste, sludge, and soil.

| Practical Field | Reaction System | Reaction Conditions | Results | Refs. |
|---|---|---|---|---|
| Actual wastewater | PVP-NZVI-Cu/PDS | [PDS] = 18 mmol/L; [catalyst] = 1.2 g/L; [TCE] = 0.15 mmol/L; pH = 3.2; t = 1 h | 99.5% of TCE removal | [103] |
| | Mag-CuO/PMS | [$Na_2SO_4$] = 0.2 mol/L; [PMS] = 2 mmol/L; [catalyst] = 0.1 g/L; pH = 7; t = 30 min; [AO7] = 0.2 mmol/L, [MB] = [RhB] = [ATZ] = 0.1 mmol/L | The removal efficiencies of AO7, MB, RhB, and ATZ were 95.81%, 74.57%, 100%, and 100%, respectively. | [104] |
| Landfill leachate | ZVINFS/rULGO/PDS | [PDS/COD] = 3; [catalyst] = 1.6 g/L; pH = 3; t = 45 min | 80.87% removal of COD and 72.38% of $NH_3$ removal | [105] |
| | $CuFe_2O_4$/PDS | [PDS] = 5 g/L; [catalyst] = 1.5 g/L; pH = 2; t = 60 min | 57% removal of COD, 71% removal of $NH_3$-N and 63% of color, respectively. | [106] |
| | $Fe_2O_3$/$Co_3O_4$/EG/PDS | [PDS] = 0.05 mol/L; [catalyst] = 0.1 g; pH = 5; t = 60 min | 90.6% removal of COD and 67.1% of $NH_4^+$-N removal | [107] |
| Biological waste sludge | ZVI/PMS | [PMS] = 24.5 mg/g TSS; [catalyst] = 260.7 mg/g TSS | SRF decreased by 83.6% | [108] |
| | UV/ZVI/PDS | [$n_{ZVI/PDS}$] = 0.6; [PDS] = 200 mg/gTSS; UV = 254 nm; pH = 6.54; t = 20 min | 64.0% decreased of CST and 78.2% decreased of SRF | [109] |
| | VTM/RH/PMS | [PMS] = 200 mg/g TSS; [VTM] = 1 g/g TSS; [RH] = 200 mg/g TSS | 94.8% reduction of CST and 63.4% of Wc | [110] |
| Soil | Fe@CF-N/PMS | [PMS] = 0.2 mmol/L; [catalyst] = 25 mg; pH = 5; t = 180 min; [FLT] = 10 mg/L | 78.12% removal of FLT | [111] |
| | Fe-Cu@BC-GM/PMS | [PMS] = 100 mg/L; [catalyst] = 100 mg/L; pH = 3; t = 120 min; [NAP] = 10 mg/L | 67.98% removal of NAP | [112] |
| | nZVI/PDS | [$n_{SMX/PDS}$] = 1/75; [catalyst] = 0.03 g/g soil; [soil/water] = 1/1 | Removal efficiencies of SMX were 87.6% (cinnamon soil), 90.6% (yellow brown earths), 80.8% (brown earths), 86.5% (black soils), and 96.1% (red earths), respectively. | [113] |

### 4.1. Actual Wastewater

The actual wastewater is complex and has a certain buffer capacity. In the treatment of real wastewater, some researchers have made some good progress by activating persulfate systems with magnetic materials. For example, Adrees et al. [103] synthesized polyvinylpyrrolidone coated Fe-Cu nanomagnetic particles (PVP-nZVI-Cu) to activate PDS to remove trichloroethylene (TCE) from groundwater, and PVP effectively prevents the agglomeration of Fe-Cu nanoparticles. However, when the catalyst dosage is 1.2 g/L and the oxidant dosage is 18 mM, the TCE removal effect of the PVP-nZVI-Cu/PDS system can reach 99.5% within 60 min, and the dechlorination rate and mineralization rate can also

reach 81.1% and 83.8%, respectively. In the system, sulfate radicals and hydroxyl radicals play a major role, whereas superoxide anion radicals play a secondary role. Overall, the system has great potential for remediation of TCE contaminated groundwater. Li et al. [104] synthesized a magnetic copper oxide catalyst (Mag-CuO) by a simple one-step precipitation method, which was used to activate PMS to treat organic pollutants in wastewater containing high salinity. In a 0.2 mol/L sodium sulfate wastewater system, the Mag-CuO/PMS system can effectively remove acid orange 7 (95.81%), methylene blue (74.57%), rhodamine B (100%), and atrazine (100%), and still maintain good catalytic activity in the system with multiple salt components. In addition, studies have shown that a singlet oxygen is the main active species in the system due to the redox cycle of Cu and Fe ($Fe^{2+}/Fe^{3+}$ and $Cu^+/Cu^{2+}$) and the hydroxylation of the material surface, because singlet oxygen is less affected by the background components in the wastewater than other radicals. At present, although magnetic materials activated by a persulfate system in the laboratory model of wastewater have made a lot of excellent progress in the study, the actual wastewater composition is complicated, so more studies that are more biased towards actual wastewater are needed.

*4.2. Landfill Leachate*

Landfill leachate is produced by rainwater or groundwater infiltration into the landfill site during the solid waste landfill process, and it has the characteristics of toxicity and low biodegradability [114]. Landfill leachate could pollute soil and groundwater because of improper treatment, which can cause potential harm to human beings and ecological environments [115]. Soubh et al. [105] used chemically expanded GO as a carrier, prepared nanofiber/super-large reduced graphene composite (ZVINFS/rULGO) by reducing $Fe^{2+}$ to ZVI supported on GO under the action of reducing agent sodium borohydride, which was used to treat landfill leachate in persulfate systems. On the one hand, the ZVINFS/rULGO/PDS system reduced the activation energy of COD and $NH_3$ by 3.8 times and 4.2 times, respectively, which was conducive to the removal of COD and $NH_3$. On the other hand, the $BOD_5$/COD of landfill leachate increased from 0.25 to 0.52, which improved the biodegradability of landfill leachate. Karimipourfard et al. [106] synthesized $CuFe_2O_4$ magnetic material for persulfate activation to treat municipal garbage leachate by a simple ultrasonic coprecipitation method. Under the strong oxidation capacity of sulfate radicals, the removal rates of COD, $NH_3$-N, and chroma in landfill leachate by $CuFe_2O_4$/PDS system are 57%, 71%, and 63%, respectively. Furthermore, Karimipourfard et al. [116] also investigated the preparation of $CuFe_2O_4$/rGO composites by loading $CuFe_2O_4$ onto reduced graphene. As the graphene carrier provides a larger specific surface area and higher catalytic activity for the oxidation reaction, compared with the $CuFe_2O_4$/PDS system under the optimal conditions, the removal of COD and $NH_3$-N by $CuFe_2O_4$/RGO/PDS system is further improved by 18.4% and 19.6%, respectively. Guo et al. [107] synthesized a peeled-graphite fixed ferrous oxide/cobalt oxide composite magnetic catalyst ($Fe_2O_3$/$Co_3O_4$/EG) by a heating-precipitation method, which was used to activate potassium persulfate to treat landfill leachate. Under optimal conditions, the removal of COD and $NH_4^+$-N reached 67.1% and 90.6%, respectively. In general, all the catalysts mentioned above have good recovery performance and reuse performance in the treatment of landfill leachate, which also confirms that PDS activation by a magnetic catalyst has a good application prospect in the treatment of landfill leachate.

*4.3. Biological Waste Sludge*

Biological waste sludge is a by-product of biological wastewater treatment, with a high moisture content. Its subsequent treatment causes significant economic and environmental burden, and improper treatment of sludge is easy to cause secondary pollution, which poses potential threats to human health and ecological environments [117,118]. Pre-treatment of biological waste sludge and reduction of its moisture content can effectively reduce the volume of sludge and reduce its transportation and disposal costs [119]. The studies showed that the spatial distribution of extracellular polymeric substances (EPS) affects the

sedimentation, biological flocculation, and dehydration performance of sludge [120,121]. Therefore, by changing the affinity of EPS to water molecules and improving the filtration capacity of sludge (i.e., reducing the specific filtration resistance (SRF) of sludge), the dehydration performance of sludge can be effectively improved, making it easy to store, transport, and treat [118]. Li et al. [108] used ZVI to activate persulfate to improve the dehydration capacity of sludge by reducing the specific filtration resistance of sludge. Under the heat assisted treatment at 50 °C, ZVI activates persulfate to produce sulfate radicals and hydroxyl radicals. Heat treatment accelerates the production of radicals and the redox process, and effectively improves the dehydration capacity of sludge (the reduction rate of SRF of biological waste sludge is 90.6%). Zhang et al. [109] pointed out that with the help of ultraviolet light, PDS activated by ZVI can also effectively degrade extracellular polymer, reducing the SRF of biological waste sludge by 78.2%. Positive ions such as $Fe^{2+}$, $Fe^{3+}$, and $H^+$ in the system can reduce the electronegativity of sludge surface and promote the agglomeration of sludge particles, thus improving the dewatering performance of sludge. In Liu's study, rice husk (RH) was used as the skeleton construction agent for the first time to activate PMS with natural (VTM) to improve the dehydration performance of biological waste sludge (VTM-PMS-RH) [110]. Under the optimal conditions, capillary suction time (CST) reduction and water content of sludge cake (Wc) were 94.8% and 63.4%, respectively. In addition, heavy metals (Cu, Zn, Cr and Pb) in waste sludge are transformed into more stable forms, reducing their leaching toxicity and thus reducing environmental risks. Yang et al. [122] used a combination of manganese ferrite/biochar (MFB)-activated PMS and tannic acid (TA) to improve the dewatering performance of biological waste sludge. The addition of TA can promote the valence state conversion of iron and manganese ($Fe^{2+}/Fe^{3+}$ and $Mn^{2+}/Mn^{3+}$), accelerate the activation of PMS, produce species with strong oxidation activity, and improve the redox capacity of the system. In addition, it is worth noting that the dehydrated sludge treated by MFB/PMS/TA is not only conducive to incineration that can generate more energy, but also can be used as a precursor system to prepare biochar with well-developed pore structure, which has a good prospect of resource utilization and is a promising and effective way to treat biological waste sludge. In general, a magnetic catalyst-activated persulfate system can effectively improve the dewatering performance of waste sludge, and has a good application status and prospects in the treatment of biological waste sludge.

*4.4. Soil*

With the development of industry and agriculture, the massive discharge of waste water and waste gas not only enters into the water, but also enters into the soil, and the existence of persistent refractory organic pollutants causes serious soil pollution [123]. Since the soil environment has limited ability to repair itself, it is very important to find effective methods to degrade organic pollutants in soil [111]. For example, Li et al. [111] prepared N-doped carbon foam-loaded Fe nanoparticle composites (Fe@CF-N) by an in-situ impregnation and unique roasting method for activating PMS to degrade fluoranthene (FLT) in soil. Fe@CF-N has a large specific surface area ($249 \text{ m}^2 \cdot \text{g}^{-1}$), which can provide more adsorption sites and active sites for the reaction. Within 180 min, the removal of FLT reached 78.12%, and sulfate radicals, hydroxyl radicals, and singlet oxygen are all involved in the degradation of FLT. Finally, the results of plant toxicity test showed that seed germination rate and root cap elongation of Fe@CF-N/PMS-treated soil were not significantly different from that of uncontaminated soil, which proved that Fe@CF-N/PMS system can be well used to treat contaminated soil. Similarly, Zhu et al. [112] synthesized Fe/Cu nanoparticle composite catalysts (Fe-Cu@BC-GM) supported by biochar/geopolymer by an impregnation-roasting method to activate PMS for remediation of soil contaminated with naphthalene (NAP). The synergistic effect between Fe and Cu (redox pair between $Fe^{2+}/Fe^{3+}$ and $Cu^+/Cu^{2+}$) effectively promoted the decomposition of PMS to produce sulfate radicals and hydroxyl radicals, which were used to degrade NAP. Moreover, the plant toxicity test also showed that the germination rate, root cap elongation, and diameter length of mung bean seeds

in the soil treated by Fe-Cu@BC-GM/PMS system were not significantly different from that of unpolluted soil, which also can prove that the persulfate/magnetic catalyst system can be used to treat contaminated soil. Zhou et al. [113] used ZVI nanoparticles to activate PDS to rehabilitate sulfamethoxazole (SMX) contaminated agricultural soils (cinnamon). Due to the weak alkalinity of the system (pH = 8.5), the active species generated during the reaction were mainly hydroxyl radicals, and the removal rate of SMX was 87.6% within 4 h. The removal rate of SMX in different types of agricultural soil (such as yellow brown soil, brown soil, black soil, and red soil) by the nZVI/PDS system is above 80%, indicating that the system has a certain positive effect on the removal of pollutants in agricultural soil. The application of magnetic catalyst-activated persulfate system in soil is still few and needs further experimental exploration.

## 5. Challenges and Perspectives

The magnetic catalyst activation of the persulfate system in wastewater containing refractory organic pollutants has achieved many good results, but there are still some limitations, and in this regard, here are some prospects to put forward:

(1) According to the magnetic species, magnetic catalysts can be roughly divided into iron-based catalyst, cobalt-based catalyst, nickel-based catalyst, and supported magnetic catalyst, and the most common is the magnetic carbon composite catalyst. However, the magnetic catalyst/persulfate system is still in the development stage and is rarely used in actual wastewater. The actual wastewater composition is complex, and in future research, we can design catalyst materials with better performance or optimize existing materials to cope with the complex composition and environment of the actual wastewater.

(2) The reaction mechanism of magnetic catalyst activation of persulfate includes the radical pathway and the non-radical pathway, which is related to the nature of the catalyst. However, some studies are not comprehensive enough for the radicals trapping experiments and EPR/ESR experiments involved in the reaction system, which may lead to a lack of comprehensive understanding of the reaction mechanism. In future research, we can combine representational means and experimental means to carry out more in-depth and comprehensive research and summary.

(3) A variety of magnetic catalysts can be used to treat refractory organic pollutants in the aqueous phase, including phenols, antibiotics, dyes, chlorinated organic pollutants, etc., but these are in the research stage of laboratory model wastewater, and there are few cases of actual wastewater treatment. In future research, the magnetic catalyst and magnetic catalyst/persulfate technology can be optimized and perfected. It is necessary to consider the switch from the laboratory scale to the middle scale, and apply the magnetic catalyst-activated persulfate system with high efficiency and low energy consumption, and make recovery of the treatment of refractory organic pollutants easy in the actual environment as soon as possible.

## 6. Summary and Outlook

In this paper, the activation mechanism of persulfate by different magnetic catalysts is introduced, including an iron-based catalyst, cobalt-based catalyst, nickel-based catalyst, and supported magnetic catalyst. The activation of persulfate by these magnetic catalysts generally involves the radical pathway and the non-radical pathway. Two methods of identifying active species involved in these two pathways (quench experiment and EPR/ESR experiment) are summarized, and the electrochemical reactions involved in electron transfer and complex formation in the non-radical pathway are also illustrated, respectively. Finally, we also list the cases in practical application. At present, there are increasing successful cases, but its practical application scope is not broad enough. In general, a magnetic catalyst has a good application prospect, which needs to be explored in the future experiments. Our aim is to make full use of a magnetic catalyst that has low energy consumption and high

efficiency, and is easy to recycle and reuse, so that a magnetic catalyst/persulfate system can be put into practical environmental treatment as soon as possible.

**Author Contributions:** K.T.: Writing—Original Draft Preparation; F.S.: Writing—Review and Editing; M.C.: Writing—Review and Editing; Q.Z.: Conceptualization; G.Z.: Conceptualization, Writing—Review and Editing, Supervision, and Funding acquisition. All authors have read and agreed to the published version of the manuscript.

**Funding:** The work was supported by the Open Project of State Key Laboratory of Urban Water Resource and Environment, Harbin Institute of Technology (ES202206), and Talents of High Level Scientific Research Foundation of Qingdao Agricultural University (6651120004) for their financial support.

**Conflicts of Interest:** The authors declare no conflict of interest.

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
