# Peer review of "A Review of Persulfate Activation by Magnetic Catalysts to Degrade Organic Contaminants: Mechanisms and Applications"

_catalysts, doi:10.3390/catal12091058_

Round 1

Reviewer 2 Report

1. The characteristics and advantages of PDS activation by magnetic materials should be clarified in Introduction.

2. The commonalities of the three mechanisms of PDS activation by magnetic catalysts should be summarized separately.

3. Line 467-468 which also confirms that magnetic catalysts have a good application prospect in the treatment of landfill leachate should be replaced by which also confirms that PDS activation by magnetic catalyst have a good application prospect in the treatment of landfill leachate.

4. Line 472-473 Please explain the significance of its existence.

5. Briefly describe the characteristics of sludge and its impact on the environment.

6. What is the purpose of the plant toxicity test? What conclusions were drawn from seed germination rate and root cap elongation?
